# Research on Design and Control Method of Flexible Wing Ribs with Chordwise Variable Camber

**DOI:** 10.3390/biomimetics11010036

**Published:** 2026-01-04

**Authors:** Xin Tao, Li Bin

**Affiliations:** School of Aeronautic, Northwestern Polytechnical University, Xi’an 710072, China; nwpu_boyd@163.com

**Keywords:** flexible wing rib, structural optimization, fuzzy sliding mode control, pneumatic muscles, chordwise variable camber

## Abstract

To improve the continuous chordwise bending performance of morphing wings, this study proposes a rigid–flexible coupled wing rib structure and its control strategy. Initially, the optimal rigid–flexible hybrid configuration was optimized via the mean camber line parameterization and genetic algorithm. For the flexible segment, topology optimization was conducted using the load path method, followed by subspace-based shape–size alternating optimization; bionic “longbow” curved beams and ‘S’-shaped substructures were adopted to enhance deformability. Biomimetic pneumatic muscles were used as actuators, and a fuzzy-adjusted PI sliding mode controller was designed to address the issue that traditional PI sliding mode controllers cannot achieve precise control under non-optimal parameters or when there is a significant difference in deformation targets. Experimental results show that when the flexible rib deflects by 15°, the three-rib wing box achieves a 30° deflection, with stresses within the allowable limit of 7075Al-T6 (540 MPa) and a deformation error of only 7.6%. For the 15° downward bending control, the adjustment time is 6.06 s, the steady-state error is 0.19°, and the overshoot is 1.8%. This study verifies the feasibility of the proposed rigid–flexible coupled structure and fuzzy PI-SMC, providing a technical reference for morphing aircraft.

## 1. Introduction

Morphing aircraft exhibit real-time adaptability to mission requirements during flight through localized or global shape adjustments, thereby enabling effective enhancements in flight performance, extending the flight envelope, and boosting flight efficiency. As the primary aerodynamic load-bearing component of an aircraft, the wing’s aerodynamic performance exerts a direct influence on the aircraft’s overall performance and flight efficiency; thus, designing a wing capable of adapting to dynamic flight states and environmental conditions is critical for maximizing these key metrics.

A morphing wing with continuous chordwise camber adjustment stands as one of the most effective approaches to realizing advanced morphing aircraft configurations. For commercial and transport aircraft, chordwise variable camber wings improve the lift-to-drag ratio across various angles of attack (AOAs), yielding fuel savings and extended range. Additionally, integrated control law design can enhance load alleviation capabilities, facilitating aeroelastic tailoring, noise reduction, and structural weight reduction [1]. For high-maneuverability military fighters, such wings further enhance lift-to-drag characteristics and flight handling qualities, reduce maneuver-induced structural loads to extend component service life, and minimize radar cross-section to improve stealth performance. Consequently, chordwise variable camber wing technology has become a focus of international research competition, recognized as a promising direction for next-generation aircraft advancement [1].

Three core aspects underpin chordwise morphing wing development: morphing wing rib design, actuator control methods, and flexible skin structures [2], among which the wing rib structural design is the most fundamental and critical. For structural design, NASA’s Dryden Flight Research Center developed the “Mission Adaptive Wing” (MAW), which was flight-tested on the F-111A aircraft [1]. The leading and trailing edges of the modified wing featured partial fiber-reinforced polymer (FRP) active surfaces, with driving mechanisms integrated into these edges to enable the real-time adjustment of wing bending and torsion angles based on flight conditions, delivering an optimal variable camber configuration. Since 2010, NASA has partnered with Boeing on the “Variable Camber Continuous Trailing Edge Flap” system [3]; focused on large transport aircraft, this project is aimed to develop a novel three-segment smooth variable camber trailing edge, actuated by shape memory alloys (SMAs) and distributed motors. Kota et al. completed the adaptive trailing edge flap design by using the flexible mechanism. The bending changeability of the adaptive trailing edge flap was tested during flight via the dynamic flight test using the test machine [4,5,6]. The German Aerospace Center proposed a step-rotatable belt–rib design [7], which enables both chordwise trailing edge camber adjustment and spanwise deformation via differential spanwise deflection. Barbarino et al. further proposed an SMA-driven variable camber trailing edge [8]; however, its deformation relies on discrete rigid-body motions, precluding strictly continuous camber adjustment. Aly et al. designed a flaperon system integrating SMAs with compliant structures, conducting finite element and aerodynamic analyses [9], but this work remains in the conceptual design stage. More recently, NASA has leveraged cellular array structures to design and fabricate ultralight morphing wings with sufficient stiffness and strength [10], while within the European “Clean Sky” initiative [11], three morphing configurations were developed for regional airliners (drooped nose, multifunctional flaps, and adaptive winglets)—all enabling real-time shape adjustments tailored to flight conditions in order to optimize aerodynamic efficiency.

Through the analysis of the structural form characteristics, the advantages of a flexible structure become more prominent. The flexible mechanism is a mechanism that changes the output displacement of the input load through its elastic deformation, storing part of the input energy in the form of strain energy. It has three key advantages: firstly, the flexible mechanism enables the structure to achieve continuous smooth deformation. Secondly, the flexible mechanism can reduce the number of required parts, shorten the processing and assembly time, and reduce the cost. Finally, the number of the flexible mechanism’s motion pairs is greatly reduced, reducing the mechanism wear and friction-induced vibration and noise [12].

Topology optimization is a critical step in wing rib structural design. Traditional topology optimization for compliant mechanisms primarily relies on the ground structure method, which facilitates manufacturing but often results in discontinuous structures and isolated elements during optimization. Additionally, the ground structure method is highly sensitive to initial meshing—different initial grid configurations can yield disparate optimization outcomes, compromising solution consistency. The load path method proposed by Lu Kerr-Jia et al. did not rely on the initial mesh, solving the structural discontinuity problem and obtaining the larger solution space [4].

In actuator selection, traditional hydraulic cylinders have been gradually replaced by advanced alternatives. Lim and Vos carried out a study on deformation drives, which was based on the piezoelectric material drive [13,14]. Sofla et al. studied shape memory alloys [15,16,17]. However, the short distance driven by piezoelectric materials is difficult to use for large aircraft. Shape memory alloys have high requirements for temperature accuracy control and are prone to fatigue problems in environments with temperature cycling. As a bionic stretching drive, the artificial pneumatic muscle has the same movement mode as human muscle. Its structure is simple and its structural characteristics include bending flexibly and being easy to miniaturize [18]. It can be installed in the complex and narrow internal space of the wing ribs, and by adopting rubber material, its own weight is significantly reduced.

For actuator control, sliding mode control (SMC) offers distinct advantages, including robustness to parameter variations and external disturbances, fast dynamic response, and no requirement for a precise system model [19]. SMC is currently widely applied in motor control, robotics, multi-joint manipulators, and inverted pendulum systems [20]. However, traditional SMC employs fixed sliding surface coefficients and gain parameters, which can degrade control performance when control objectives change significantly. The PI sliding mode control method mentioned in Article [21] struggles to achieve precise control with the same control parameters under varying target angle requirements. Fuzzy logic enables the dynamic adjustment of SMC parameters based on real-time error and error rate, forming an adaptive mechanism that mitigates the risks of overly conservative or insufficient parameter settings while enhancing system robustness to uncertainties and disturbances [22].

Motivated by these challenges, this study integrates compliant mechanisms into the design of chordwise variable camber wing ribs. A concise overview of the optimization method for the optimal variable camber trailing edge configuration is first presented. Subsequently, the load path method is employed for preliminary topology optimization, followed by alternating subspace optimization of shape and size to derive the optimal topological configuration. With pneumatic muscles as actuators and the downward bending angle of the flexible trailing edge as the control objective, a fuzzy-tuned PI-SMC controller is designed. Finally, a control experimental system is constructed to verify the deformation capability of the flexible trailing edge and to validate the feasibility of the proposed control strategy.

## 2. Materials and Methods

### 2.1. Materials

Wing rib material: 7075Al-T6 alloy (allowable stress: 540 MPa), selected for its high strength-to-weight ratio (suitable for aerostructures).

Actuator: Pneumatic muscle (FESTO DMSP-20, sourced from Festo AG & Co. KG, Esslingen, Germany), with metal fixed ends, a rubber cylinder (central portion), and metal fiber wrapping; operating pressure range: 0–6 atm; and weight: 0.05 kg (lightweight for wing integration) [18].

Test equipment: XFoil (2D airfoil aerodynamic calculation, version 6.96), Simulink (control system simulation, within MATLAB R2014a), strain gauge (stress monitoring, BE120-3AA-P200, Zhonghang Electronic Measuring Instruments Co., Ltd., Hanzhong, China), high-speed camera (deformation measurement, model D850, Nikon Corporation, Tokyo, Japan).

### 2.2. Overall Configuration Optimization of Variable Camber Wings

The flaps installed at the trailing edge of the wing typically constitute 25–35% of the overall wing profile chord length. Using the wing of a typical medium-sized transport aircraft as the reference design size, the total chord length of the wing has been established to be 3.33 m, with a trailing edge variable camber section measuring 1 m, accounting for 30% of the chord length of the trailing edge flap. Additionally, based on the existing flap system, the target deflection angle for the trailing edge of the chord variable camber section has been determined to be 30°. The NACA0012 airfoil was used as the basic airfoil for both the design and its optimization.

Structural design usually consists of three parts: input point, structure, and output point. The output point set of the chordwise variable camber trailing edge is the outer rib structure contour. Furthermore, the input point is the driving position, which can be divided into centralized single-point drive and distributed drive. The centralized single-point driving method has defects such as relatively single-load transfer path, easy to produce local stress concentration, uneven deformation, and poor smoothness. Further analysis of the continuous variable bending demand in the mean camber line of the airfoil trailing edge has shown that the wing trailing edge deformation is similar to that of the cantilever beam under load. Moreover, the initial and tip arc deformation segments in the cantilever beam trailing edge show similar slope distribution [23].

Considering the elastic-allowable deformation of the wing rib material and distributed drive form, a hybrid structure coupling deformable wing rib design was proposed, as shown in Figure 1. In the variable bending section of the wing trailing edge, structure types were arranged alternately in three sections. They were rigid segment, flexible segment, and follow-up segment, which accounted for 30%, 40%, and 30% of the total wing trailing edge.

The optimization of the bending configuration is not the focus of this article, but to ensure the readability and coherence of the text, we will briefly introduce the process of configuration optimization here.

To describe the diversity and continuity of the downward bending of the trailing edge flaps of an aircraft wing, the variable camber configuration can be equivalently described using the mean camber line. Thus, a relatively intricate issue concerning the parameterization description of airfoils can be converted into a problem of curve parameterization description. We utilize cubic spline equations to delineate the mean camber line of the airfoil.

In Figure 2, the six points (m0 to m5) are located on the mean camber line of the airfoil, where m0 is the leading edge point of the variable wing camber section and m5 is the trailing edge point. Furthermore, m1 and m4 are the rigid–flexible junction points, while m2 and m3 are points with x-coordinates of 40% and 60% of the trailing edge chord length, respectively. Moreover, points m2 and m3 are located on the middle flexible segment arc [21].

The geometric relationship was obtained by introducing the coordinates of each point into Formula (1). The y-coordinate of each point was selected as the optimization variable.(1)$$\left\{\begin{array}{l} y=ax^{3}+bx^{2}+cx+d\\ \int_{x_{\mathrm{start}}}^{x_{end}}\sqrt{1+{\left(\frac{dy}{dx}\right)}^{2}}dx=l\\ y_{\mathrm{start}}=0\\ y_{end}-y_{\mathrm{start}}=\left(x_{end}-x_{start}\right)\cdot \mathrm{tand}\left(\theta_{2}\right) \end{array}\right.\to \left\{\begin{array}{l} x_{m0}=y_{m0}=0\\ y_{m1}=x_{m1}\cdot tan(\theta_{1})\\ f^{\prime }(x){|}_{x=x_{m1}}=3a\cdot x_{m1}^{2}+2b\cdot x_{m1}+c=tan(\theta_{1})\\ y_{m1}=a\cdot x_{m1}^{3}+b\cdot x_{m1}^{2}+c\cdot x_{m1}+d\\ y_{m2}=a\cdot x_{m2}^{3}+b\cdot x_{m2}^{2}+c\cdot x_{m2}+d\\ y_{m3}=a\cdot x_{m3}^{3}+b\cdot x_{m3}^{2}+c\cdot x_{m3}+d\\ y_{m4}=a\cdot x_{m4}^{3}+b\cdot x_{m4}^{2}+c\cdot x_{m4}+d\\ \frac{\left(y_{m5}-y_{m4}\right)}{\left(x_{m5}-x_{m4}\right)}=f^{\prime }(x){|}_{x=x_{m4}}=3a\cdot x_{m4}^{2}+2b\cdot x_{m4}+c\\ \int_{x_{m1}}^{x_{m4}}\sqrt{1+{\left(3a\cdot x^{2}+2b\cdot x+c\right)}^{2}}dx=l_{m1m4}\\ \sqrt{{\left(y_{m5}-y_{m4}\right)}^{2}+{\left(x_{m5}-x_{m4}\right)}^{2}}dx=l_{m4m5} \end{array}\right.$$
where $\left(x_{\mathrm{start}},y_{start}\right)$ are the initial mean camber line deflection point coordinates, $\left(x_{end},y_{end}\right)$ are the trailing edge coordinates of the mean camber line, l represents the variable curvature section arc length, and $\theta_{2}$ is the equivalent deflection angle of the variable curvature section.

The maximum lift-to-drag ratio coefficient was selected as the evaluation function for finding the optimal variable bending configuration [21], as follows:(2)$$S=\max(\frac{C_{L}}{C_{D}})$$

In Figure 2, point m1 serves as the connection between the rigid rotation segment and the flexible segment. Given the relatively independent driving methods of these two segments, the tangent slope at this point will vary, and a certain degree of deviation should be allowed. However, considering the continuity of the overall airfoil camber change, it is stipulated that the sudden change in the deflection angle caused by this deviation should be less than 10% of the overall deflection angle. The deformation target angle is set at a 30° downward deflection of the trailing edge of the wing, which is the maximum design deflection angle corresponding to the takeoff/landing state of the aircraft. The lift-to-drag ratio is calculated using the XFoil two-dimensional airfoil aerodynamic calculation software. Additionally, the flow field parameters are set to the takeoff/landing conditions, as shown in Table 1.

The coordinates in the optimization variables are updated using genetic algorithms. The optimization process was established using the following constraints: the material strength limitation, maximum allowable equivalent chord deflection angle of 15° for the flexible segment, and the follow up segment. The optimal coordinates obtained from each generation of optimization are normalized. Then, the airfoils are generated and the lift-to-drag ratio is computed. Finally, the generated optimal chordwise variable camber airfoil is compared with an airfoil having a mean camber line in the form of a cantilever beam.

### 2.3. Structural Optimization of the Flexible Segment

The structural optimization of the wing rib trailing edge structure design is necessary to ensure the smooth wing rib structure and continuous deformation. The rigid deflection segment is driven by servo motor and linkage mechanism. Making the design of the flexible segment configuration and driving mode is challenging. When the structural form is unknown, the topology optimization method can generally be used to design the structural configuration.

The ground structure method is often used for the topology optimization of discrete structures. However, due to its dependence on initial nodes and the corresponding full-base structure, there may be a risk of local optimal solutions and free elements that are disconnected from the structure. The load path method is similar to the ground structure method in the initial layout; however, its optimization approach is different. The characteristic design domain points are connected by the load path transfer form. As the load path method directly tracks the force transmission path within the structure, making it clear how loads are transferred from input points to output points, it provides an intuitive representation of the structural loading logic. In contrast to the ground structure method, this method can drastically reduce the number of redundant elements and improve optimization efficiency. The diversity of the topology solutions is controlled by both the position and number of intermediate points. Firstly, the appropriate number of nodes is arranged in the design domain, and the initial design-domain topological structure is given through the full base configuration. Then, the load path is parameterized and discretized to create path forms, different from the load input point, to the output point. Subsequently, appropriate design variables, such as path number and path weight, are used to find the optimal path from the load input to the output point. The optimization algorithm and finite element method are used for calculation. Finally, all the structures covering these paths are combined, obtaining the final topology.

Taking the flexible segment of the trailing edge variable camber rib structure as an example, the topology optimization design domain is shown in Figure 3.

According to the flexible segment deformation requirements, output points are evenly distributed on the upper and lower wing rib surfaces, with a total of six output points (points 1, 2, 3, 4, 5, 6). There is one input point, and, since its position is at the lower wing rib edge, it is both the input and output point(point 2). The middle node is an important segment of the load path method. Different topological shapes are obtained by varying the number and location of the middle node. Simultaneously, the middle node is also a bridge connecting the elements. Therefore, determining the number of middle nodes and their locations is a rather important task. Considering the completeness of the topological shape results, the optimization calculation efficiency, and the processing feasibility, it is reasonable to select 4–10 intermediate nodes. In this article, five intermediate nodes were used (points 7, 8, 9, 10, 11). Points 12, 13, and 14 are fixed, with point 14 having freedom of movement along the X direction.

After determining the initial design area topology, each load path must be both determined and parameterized. According to the load path method’s definition, the load path is the path used to transfer the load between the input and output point. Once the topological shape is determined, each element point is calculated accordingly, and the load path associated with it can be determined. The load path method’s algorithm flowchart is given in Figure 4.

The load path method’s design variables include the load path sequence, the maximum length of the load path, the material Young’s modulus, the input load size and direction, the number and location of element points, and the element section size. In this paper, the structure is a single-input multi-output system. Considering the modeling and optimization complexity, the length weight of the design variable load path was set to one. Furthermore, the structure’s section width was taken as constant and the 7075Al-T6 alloy was selected as the material. Therefore, in the design of the variable trailing edge of the flexible wing based on the load path topology optimization method, the design variables were as follows: load path sequence and the position of the intermediate node.

In Figure 3, point 1 is the driving load input point, while points 1–6 are the output points located on the upper and lower surfaces of airfoil. Points 12, 13, and 14 are the connection points of the rigid segment and the flexible segment. In order to reduce the stress of flange during deformation, the free degree of point 14 in X direction is released in the finite element model. Firstly, topology optimization based on the load path method requires the identification of the force transfer path from the input point to the output point. Therefore, Dijkstra’s Greedy algorithm is employed to determine the shortest path from the input point to each output point, in order to initially screen effective load paths [24]. Next, the first *K* load transfer paths from the input point to each output point were solved via the shortest path.

In this algorithm, the path matrix had to be solved first. The requirement is that, if the input point is directly connected with the output point, the element is equal to the path weight. The weight of each path in this paper is one. If the input point and the output point are connected through the intermediate point, the element value is infinite. Therefore, the path matrix of the load path method can be given as follows:(3)$$\begin{array}{rr} & \begin{array}{cccccc} 1\;\; & 2\;\; & 3\;\; & 4\;\; & 5\; & 6\;\;\; \end{array}\\ \begin{array}{c} 1\\ 2\\ 3\\ 4\\ 5\\ 6 \end{array} & \left[\begin{array}{cccccc} 0 & 1 & \infty & \infty & 1 & \infty \\ 1 & 0 & 1 & \infty & \infty & \infty \\ \infty & 1 & 0 & 1 & \infty & \infty \\ \infty & \infty & 1 & 0 & \infty & \infty \\ 1 & \infty & \infty & \infty & 0 & 1\\ \infty & \infty & \infty & \infty & 1 & 0 \end{array}\right] \end{array}$$

Using the first shortest path from the input point to each of the output points, the first *K* path sequences from the input point to output points can be obtained via extended calculation. Through analysis, when *K* = 4, all paths have covered all the initial base structures. Hence, when *K* = 4, each path is used to the subsequent calculation. The structure numbers in the first four paths from the input point to the output point are shown in Table 2.

Based on the above-provided path numbers, it can be determined whether each beam structure in the design domain is called by the path. If it is, it remains in the finite element model; if not, the beam is deleted. The minimum square difference between the actual variable bending configuration and the target configuration was selected as the space optimization process objective function:(4)$$\min f(x)=\sum_{i=1}^{n}\sqrt{{\left(x_{i}-{\bar{x}}_{i}\right)}^{2}+{\left(y_{i}-{\bar{y}}_{i}\right)}^{2}}$$

Combined with the finite element model calculation, design variables were iteratively updated according to the objective function. Finally, the topology optimization results of the flexible wing trailing edge were obtained. The internal structure of the flexible structure is shown in Figure 5, in which the beams represented by thin lines are the beams abandoned in the optimization process.

Aiming to ensure that the actual deformed outer contour will further approach the target contour curve, it is necessary to further optimize the results of topological optimization. In shape optimization and dimensional optimization, the node positions and shape of beam elements and the cross-sectional dimensions of the elements are the optimization variables.

Therefore, the design idea of equal strength variable cross-section curved beam was introduced. The ancient longbow bionic structure and ‘S’ substructure were used to replace the straight beam structure used in the initial topology model. Table 3 indicates that when using a force form similar to a wing rib as the input load, the longbow structure displays enhanced stability and greater susceptibility to deformation. Meanwhile, in order to improve the structural deformation capability, Figure 6 presents a comparison between the load displacement curves of a straight beam and an ‘S’-shaped substructure. It was found that the ‘S’ substructure displayed a greater deformation displacement given the same radial load, making it easier for the structure to approach the target contour curve. Therefore, the multiple short beams connecting the input point and the fixed point can be replaced with the longbow-type constant strength variable section curved beam. Consider beams 1-7, 7-8, 8-10, and 10-12 in Figure 4 as a whole arched beam. The short beam substructure in the upper and lower flanges can be replaced with the ‘S’-type substructure. Replace beams 10-4, 8-3, 11-6, and 11-5 retained through topology optimization in Figure 4 with ‘S’ type beams, respectively. At the same time, replace beams 8-11, beams 11-9, and beams 9-1 that connect the intermediate points with curved beams.

The structural size and the node location significantly affect the stress level and variable bending configuration during the structural deformation. Therefore, the coupling of the two was introduced during the optimization. For such problems, separate subspaces for various optimization variables are created using their respective evaluation functions, followed by alternating optimization iterations. This is illustrated in Figure 7.

In the node location optimization subspace, the geometric position of each node was used as the optimization variable. The degree of fit between the actual variable camber configuration and its optimal value of the flexible wing rib structure was used as the evaluation criterion (under the given driving force), as shown in Formula (4).

In the subspace dedicated to size optimization, structural deformation is primarily facilitated by the main beam connecting the input point to the fixed point, whereas the short beam structures linking the main beam to each output point serve to adjust the deformation of the overall structure. Consequently, the stress distribution within each beam structure during the deformation process holds significant importance. Hence, leveraging the equal strength beam theory, size optimization has been conducted on the variable cross-section main beam that spans the entire flexible wing rib structure. In engineering applications, the complexity of structures makes it impractical to maintain consistent stress values across every beam element section. Consequently, during the size optimization of the wing rib’s primary load-bearing beam structure, this paper focuses on selecting key nodes and obtaining their stress values. By updating the optimization variables, we aim to minimize the deviation between the stress values of each section and the target stress values set according to the equal strength beam theory. This is shown in Formula (5).(5)$$\min g(x)=\sum_{j=1}^{m}\left|\sigma_{\mathrm{target}}-\sigma_{j}\right|$$$$S.T:\;\max\sigma <\sigma_{yield}$$

The rectangular beam cross-section was selected in the structure. When the bending of the wing rib flexible segment structure changes in the chord direction, the beam element thickness H will significantly affect the structural stress level. The main penetrating beam can be divided into the front section, middle section, and tail sections with two key nodes as dividing points. Each beam element was continuously arranged with respect to the sectional points W1–W11. The schematic diagram of sectional point arrangement is shown in Figure 8.

The genetic algorithm was used for optimization in different subspaces. In each iteration, the finite element analysis of the flexible variable bending process of the structure was carried out. Next, the established evaluation function could be used to determine whether the structural design meets the requirements.

After numerous rounds of optimization and iteration, the expressions for the specific dimensions of the thickness H in the front, middle, and rear sections of the main beam were obtained. Upon normalization, the results are presented as shown in Equation (6):(6)$$h(x)=\left\{\begin{array}{c} 13.396x^{3}-19.094x^{2}+5.614x+3.569,x\in [0,1]\\ 6.887x^{3}-4.818x^{2}-2.675x+3.486,x\in [0,1]\\ -8.818x^{3}+20.343x^{2}-13.156x+2.881,x\in [0,1] \end{array}\right.$$

Finite element modeling was performed on the optimized wing rib structure, utilizing 7075Al-T6 as the material. This material boasts an elastic modulus of 72 GPa, a Poisson’s ratio of 0.33, and a density of 2810 kg/m^3^. In the modeling process, beam elements were applied to the flexible and follow-up segments, whereas shell elements were used for the rigid rotational segment. Given that this paper primarily investigates the deformation of the flexible segment, the boundary conditions were configured accordingly: the rigid rotational segment was partially fixed, and two out of the three connection points between the flexible and rigid segments were fixed. The connection point at the lower edge of the wing rib was allowed freedom in the X direction to alleviate internal structural stress. When the connection point at the lower edge was also immobilized, indicating that the flexible structure was rigidly connected to the rigid segment at three points, under such boundary conditions, a significantly larger driving force was required to attain the target deformation at the endpoints of the structure. Concurrently, the maximum stress near the connection at the lower edge of the rib and within the ‘S’-shaped substructure located at the lower portion of the rib substantially surpasses the material’s permissible stress, leading to structural damage. The finite element model underwent a stepwise mesh refinement from coarse to fine, with computations carried out on three distinct mesh densities. The results showed that the maximum stress and deformation were both below 2%, confirming mesh convergence, as detailed in Table 4. Consequently, the second mesh density was ultimately chosen for the calculations, as depicted in Figure 9.

### 2.4. Control System Design

The hybrid structure variable camber wing rib in this paper needs distributed driving. The rigid segment uses a servo motor to drive a linkage mechanism for precise deflection control. The flexible structure, the design’s focus, with a longbow-like variable cross-section beam as its main deformation part, uses pneumatic muscle (Figure 10), an actuator with simple structure, featuring light weight, easy bending and miniaturization. The pneumatic muscle features metallic fixed ends and an air inlet device, while its central portion consists of a rubber cylinder wrapped with metal fibers. Its principle facilitates bow-drawing driving, and is suitable for the wing’s complex narrow space.

Figure 11 depicts the control process for driving and controlling the flexible structural using pneumatic muscles [21].

Due to the abovementioned control process and controller design, the mathematical modeling of the pneumatic muscle driving force and the wing rib’s flexible segment structure was needed.

Establishing a mathematical model for the driving force of pneumatic muscle is the primary method used to quantify its driving capability, and it also serves as the foundation for constructing a simulation model of the variable bending drive control system for the flexible wing rib structures [25]. Pneumatic muscle was used as the research object, as shown in Figure 12.

Here, *m* is pneumatic muscle quality, *L* represents the real-time length of the pneumatic muscle, *F*_1_ is the elasticity of pneumatic muscle rubber sleeve, *F*_2_ is the force load, and *F* is the pneumatic muscle contraction.

Based on the derivation of the dynamic equations during the pneumatic muscle actuation process in reference [21], we obtain the expression for the driving force, as follows:(7)$$F=\frac{p-p_{0}}{4\pi n^{2}}\left(3L^{2}-l^{2}\right)$$
where p is the internal pneumatic muscle pressure, $p_{0}$ is the standard atmospheric pressure, l is the length of the single fiber, and n is the number of wraps around the fiber.

The first step in establishing the simulation model of the wing rib’s flexible segment is to identify the structural parameters, as shown in Table 5.

After analyzing the deformation characteristics of the flexible rib structure, its dynamic model was established using the Lagrangian method, as follows [21,26]:(8)$$M\ddot{q}+C\dot{q}+G\left(q\right)+K(q)=\tau -\tau_{d}$$
where M is the structure moment of inertia, *C* is the structural bending damping, *G* is the rotational torque generated by structural gravity, *K* is the rotational stiffness of the structural equivalent joint, and, finally, τ and $\tau_{d}$ are the control torque and external disturbance, respectively.

In practical applications, physical parameters and disturbances are often inaccurate, and flexible wing rib parameters vary with camber changes. Therefore, system modeling errors must be considered. To obtain an accurate dynamic model, the coefficient terms in the dynamic Equation (8) are divided into deterministic and error components (referred to as the nominal model). Using the sliding mode variable structure control method, the controller expression is obtained as follows [21]:(9)$$\tau =\tau_{m}+K_{p}r+K_{i}\int rdt+\tau_{r}$$
where *K_P_* is the proportional coefficient, *K_i_* is the integral coefficient, and $K_{p}>0,\;K_{i}>0$, *r* is the sliding surface function. $\tau_{m}$ is the control rate based on the nominal model and $\tau_{r}$ is the robustness term required to mitigate the system modeling error and external interference; the expression is as follows:(10)$$\tau_{m}=M_{0}(q){\ddot{q}}_{r}+C_{0}(q,\dot{q}){\dot{q}}_{r}+G_{0}(q)+K_{0}(q)$$(11)$$\tau_{r}=K_{r}sgn(r)$$
where $M_{0},\;C_{0},\;G_{0}$, and $K_{0}$ are deterministic components of the coefficient of the nominal model and *q* is the actual angle signal. The expression for ${\dot{q}}_{r}$ is as follows:(12)$${\dot{q}}_{r}=r(t)+\dot{q}(t)$$

In Equation (11):(13)$$K_{r}>E^{\prime }+\tau_{d}$$(14)$$E^{\prime }=E_{M}{\ddot{q}}_{r}+E_{C}{\dot{q}}_{r}+E_{G}+E_{K}$$
where $E_{M}$, $E_{C}$, $E_{G}$, and $E_{K}$ characterize the error components of coefficients of the nominal model.

A simulation platform for a variable bending controller of flexible structures was constructed using Simulink. Taking the bending of a flexible structure by 5° as an example, multiple sets of control parameters (*K_P_* is the proportional coefficient, *K_i_* is the integral coefficient, and Λ is the sliding mode surface coefficient) were compared. It was determined that when $K_{p}=2,\;K_{i}=0.5,\;\Lambda =20$, the system had the shortest adjustment time, as well as the smallest steady-state error and overshoot (See Table 6 and Figure 13) [21].

The sliding mode variable structure control algorithm has the advantages of quick response and strong robustness. However, it is difficult for the same set of parameters to achieve the best control effect at all possible target angles. Meanwhile, when it is difficult to obtain the optimal parameters, the control effect will also be reduced. To tackle the aforementioned issues, we can draw upon relevant experience obtained from early experimental results and incorporate fuzzy control to fine-tune the parameter settings within the sliding mode control algorithm [27]. Finally, all the target angles within the scope could be ensured to have good controlling effect. The simulation process is shown in Figure 14.

Adopting a two-dimensional fuzzy controller, proportional coefficient $K_{p}$ and integral coefficient $K_{i}$ are directly adjusted through the fuzzy control rate. Error e and its rate of change $\dot{e}$ were set as the input variables for the fuzzy controller.

Firstly, the definition of fuzzy sets is as follows: PB means positive big; PS means positive small, ZO means zero, NS means negative small, and NB means negative big.

Secondly, according to the fuzzy control principle, e and $\dot{e}$ are defined as the fuzzy control input, and $\Delta K_{p}$ and $\Delta K_{i}$ as the fuzzy control output. They can be expressed as in Equation (15).(15)$$\left\{\begin{array}{l} e=[NB,NS,ZO,PS,PB]\\ \dot{e}=[NB,NS,ZO,PS,PB]\\ \Delta K_{p}=[NB,NS,ZO,PS,PB]\\ \Delta K_{i}=[NB,NS,ZO,PS,PB] \end{array}\right.$$

Thirdly, confirmation of fuzzy control rule and reasoning algorithm: The used fuzzy rule works as follows: If e is A and $\dot{e}$ is B, then $\Delta K_{p}$ is C and $\Delta K_{i}$ is D. The Mamdani reasoning method is employed for reasoning algorithm, which is widely used in fuzzy control. The rules of fuzzy control are shown in Table 7.

Fourthly, according to the fuzzy control principle, the domain of input parameters e and $\dot{e}$, and the output parameters $\Delta K_{p}$ and $\Delta K_{i}$ are defined as in Equation (16). The membership function of all the parameters in the fuzzy control is confirmed to be a triangular function.(16)$$\begin{array}{l} e\in [-0.5,0.5];\dot{e}\in [-10,10]\\ \Delta K_{p}\in [-5,5];\Delta K_{i}\in [-10,10] \end{array}$$

### 2.5. Experimental Setup

Based on the structural optimization results, the 7075Al-T6 alloy material was processed and tested for variable bending capacity. At the same time, after referring to the numerical calculation results, real-time strain monitoring was carried out at typical locations within the structure. A test system consisting of drive, control, and measurement modules was constructed; a 15° downward bending test was conducted; and the adjustment time, steady-state error, and overshoot were recorded. Simultaneously, a representative wing box, incorporating a three-rib structure and a sliding skin configuration, was constructed, and its capability for hybrid structure deflection was evaluated.

The experimental system comprises three components: the drive system, the control system, and the acquisition system.

The drive system comprises an air source and a DMSP-20 pneumatic muscle tendon produced by FESTO. The control system incorporates a proportional pressure regulating valve, a dSPACE semi-physical simulation system, and a control computer. The acquisition system includes a static strain gauge, an m + p signal collector, and a laser displacement sensor. The measurement tolerance of the laser displacement sensor used for precise angle calculation is ±0.7% of the full scale. The strain coefficient tolerance of the strain gauge is ±1.0%. The schematic diagram of the experimental system is shown in Figure 15. The numbers in the figure indicate the positions and numbers of the strain gauges.

## 3. Results

### 3.1. Optimal Variable Camber Configuration

The coordinates in the optimization variables are updated using genetic algorithms. The optimization process was established using the following constraints: the material strength limitation, maximum allowable equivalent chord deflection angle of 15° for the flexible segment and the follow up segment. The optimal coordinates obtained from each generation of optimization were normalized. Then, the airfoils are generated and the lift-to-drag ratio computed. Finally, the generated optimal chordwise variable camber airfoil is compared with an airfoil having a mean camber line in the form of a cantilever beam, as shown in Figure 16. As illustrated in the figure, when subjected to an identical equivalent downward bending angle, the hybrid trailing edge demonstrates a more pronounced downward deflection and a more seamless curvature distribution compared to the trailing edge of the cantilever beam.

### 3.2. Structural Optimization Results

The topological structure obtained from Figure 5 is further optimized. An iterative method that couples substructure size and node position in alternating subspaces is employed. To ensure that the optimal variable camber configuration is achieved while maintaining structural stress within the allowable stress range of the material, the final iteration of the alternating iteration process is dedicated to size optimization. The deformation and stress distribution of the optimal flexible rib structure are shown in Figure 17. Figure 17a depicts the bending deformation of the flexible and follow-up segments, with a deflection of 172 mm at the trailing edge wing tip, corresponding to the 15° downward bend of the trailing edge structure of the wing rib. In Figure 17b, the stress throughout the entire structure remains below the material’s allowable stress of 540 MPa, with the maximum stress occurring at the junction of the ‘S’-shaped substructure on the upper left side and the bionic arched main beam, reaching a value of 506 MPa. Chamfering during the processing of the test piece can effectively reduce the stress concentration at this location. As the main deforming structure, the bionic arched main beam exhibits its maximum stress in the middle of the beam, reaching a value of 447 MPa.

Based on the optimization results of the variable trailing edge flexible segment, which were combined with the design of the rigid rotation section, the structural form of the hybrid trailing edge was obtained (see Figure 18).

### 3.3. Deformation Capacity Test

Based on the structural optimization results, the 7075Al-T6 alloy material was processed and tested for variable bending capacity. At the same time, after referring to the numerical calculation results, real-time strain monitoring was carried out at typical locations within the structure.

Through observation and calculation, under the 6.5 V driving voltage, the internal pressure of the pneumatic muscle was 3.91 atm, while the equivalent deflection angle of the wing rib flexible segment structure reached 15°. Figure 19 displays experimental photographs of a flexible segment rib structure bending downwards by 15° under the actuation of pneumatic muscles [21], as well as a wing box structure comprising three ribs and a sliding skin structure [28], which bends downwards by 30° when driven collectively by both servo motors and pneumatic muscles. In the main through-beam and its substructure, the maximum tensile stress level appeared in the middle of the main beam deformation section; its value was 409.15 MPa. The maximum compressive stress level was found in the root substructure of the lower flange edge and was equal to −467.61 MPa. It should be added that the allowable stress of the alloy material is 540 MPa.

While verifying the variable bending ability of the flexible wing rib section, the marker point coordinates under the actual structural target deflection angle were fitted to obtain the configuration diagram of the actual structure’s variable bending. The diagram was compared to the configuration of the target variable bending, as shown in Figure 20. The error between the two structures was only 7.6%, which was mainly attributed to the difficulty in machining the variable bending beam in the rib model specimen and assembling the test system.

### 3.4. Control Performance Test

The PI sliding mode variable structure control method can achieve good control effects under optimal control parameters. When optimal control parameters cannot be precisely obtained, or when the control objectives undergo significant changes, the PI sliding mode variable structure control may encounter issues such as long adjustment time and excessive overshoot that may damage the structure. Therefore, a fuzzy control module is utilized to correct the control parameters. Table 8 presents the dynamic response results of the system under non-optimal control parameters, with or without the application of fuzzy control correction. Figure 21 provides a comparison of the response curves. Table 9 presents the dynamic response results of the system when significant changes occur in the control target, with or without the application of fuzzy control correction. Figure 22 provides a comparison of the response curves.

In practical engineering applications, if the parameters of sliding mode variable structure control are difficult to obtain or incorrectly set, it will seriously affect the control effect and even cause structural damage. Taking the 5° deflection target angle of the flexible segment as an example, as shown in Table 8 and Figure 21, when non-optimal control parameters are employed, the flexible structure experiences an overshoot of 16.41%. Nevertheless, after dynamically adjusting the proportional and integral coefficients in sliding mode variable structure control through a fuzzy control algorithm, the overshoot decreases to 3.00% with the same set of control parameters, and the adjustment time is reduced from 3.11 s to 2.79 s.

It can be seen from Table 9 and Figure 22 that when the deflection target of this flexible segment was at its maximum value at a 15° angle, the optimal parameters determined under sliding mode control simulation, i.e., $K_{p}=2,\;K_{i}=0.5,\;\Lambda =20$, were as follows: its control time was 6.31 s, the overshoot was 6.21%, and the step response peak was 15.93°, which was higher than the deflection angle designed in flexible segment. It could potentially damage the structure. When proportional coefficient $K_{p}$ and integral coefficient $K_{i}$ were dynamically revised using the fuzzy control algorithm, the control time was only 1.23 s, the overshoot was 2.33% and the step response peak was 15.35°, which played a certain role in protecting the structure.

As can be seen from Figure 21 and Figure 22, the sliding mode variable structure control method, dynamically revised by the fuzzy control, met the requirements of each target angle, which could achieve high quality control.

While verifying the deformation capability of the flexible wing rib, the dynamic performance of the control method was tested. Using the optimal control parameters obtained from numerical simulation, the response diagram of structural deformation was obtained when the flexible wing rib structure was deflected by 15° and the trailing edge tip of the rib was deflected downward by 171.3 mm, through a fuzzy-modified sliding mode variable structure control method. In the experiment, real-time feedback on the angle was acquired by measuring the displacement at the root of the wing’s lower edge with a laser displacement sensor, followed by subsequent calculations. Consequently, Table 10 illustrates the correlation between the driving air pressure, this displacement, and the actual deformation angle throughout the control process. The performance is evaluated by reading and calculating the three performance evaluation indexes of the step response process—adjustment time, steady-state error, and overshoot. The step response of the control process is shown in Figure 23. The response time was 6.06 s, the steady-state error was 2.31 mm, the corresponding rib deflection angle was 0.19°, and the overshoot was 1.8%.

## 4. Discussion

This study elaborates on the configuration design, structural optimization, and actuation control of a hybrid chordwise variable camber wing rib. The findings demonstrate that integrating a bio-inspired optimized flexible structure, an actuation system, and a fuzzy sliding mode variable structure control strategy enables substantial, precise, and continuous wing deformation.

A pivotal challenge in the design of chordwise-bending morphing airfoils lies in achieving a seamless, continuous aerodynamic surface, in contrast to the discrete displacements typically observed in conventional flaps or early morphing concepts, such as stepped rotating ribbons [7]. This study utilizes flexible mechanisms to provide such continuous deformation, which aligns with the goals of advanced projects like NASA and Boeing’s VCCTEF [3] and Kota’s flexible trailing edge [4,5,6]. However, while many studies on flexible structures employ ground structure methods, variable density methods, or other techniques for topology optimization, the load path method [4] adopted in this study presents distinct advantages. By directly tracing the force transmission path, this method resolves the issues of structural discontinuities and free elements that often arise in ground structure methods due to grid dependency. Further enhancements were achieved by incorporating biomimetic ‘longbow’-shaped curved beams and ‘S’-shaped substructures. As illustrated in Table 3, compared to straight beams, the longbow beams demonstrate superior endpoint displacement under load, while the ‘S’-shaped beams exhibit a more favorable load displacement curve for flexible deformation (Figure 6).

In the selection of actuators, although shape memory alloys have been extensively researched [8,9,15] and exhibit high power density, they face challenges in precise temperature control and potential fatigue under thermal cycling. Piezoelectric actuators, as used in other studies [13], typically offer limited stroke and are not suitable for large-scale wing deformation. As biomimetic actuators, pneumatic muscles [18] possess significant advantages, such as a simple structure, light weight, high force-to-weight ratio, high energy density, and good compliance. Additionally, they are minimally affected by temperature and can be customized to different specifications according to varying needs. They are highly suitable for integration into the limited space of wing ribs, effectively addressing practical constraints in wing installation.

In the field of control strategy research, sliding mode control has found widespread application in areas like robotics, owing to its independence from model parameter identification and robustness [19,20]. Nevertheless, as indicated in the literature [21], a significant drawback of SMC is its performance deterioration under non-optimal parameters or fluctuating operating conditions. The fuzzy-modified PI-SMC proposed in this study effectively overcomes this limitation. The fuzzy inference system dynamically adjusts control parameters in response to real-time error and its rate of change, thereby compensating for the limited adaptability of fixed-parameter controllers. When employing non-optimal parameters for a 5° deflection, fuzzy logic reduced the overshoot by 81.7% (Table 8, Figure 21). For a maximum deflection of 15°, the adjustment time was reduced by 85.0%, with the overshoot being controlled at a mere 2.33% (Table 9, Figure 22), ensuring high-performance control across the entire operating range. This is of paramount importance in preventing structural damage caused by excessive overshoot during large-scale deformation processes.

In the experiment, there was a 7.6% error between the actual deformation contour and the target deformation contour, which is acceptable for the prototype. However, it poses challenges in manufacturing complex geometric shapes and system assembly. Additionally, it is necessary to incorporate aerodynamic loads into subsequent research to analyze the load-bearing capacity of the wing. Furthermore, a coupled aerodynamic–structural–servo control model should be established to study the aeroelasticity of the wing and discuss the impact of changes in the stiffness of flexible structures on the aeroelastic performance of the wing.

In summary, this study establishes a viable framework for chordwise morphing wings. By integrating structural optimization with an adaptive fuzzy sliding mode control strategy for pneumatic muscle actuation, it addresses some of the limitations inherent in previous methods, particularly in achieving robust and precise control over large-scale morphing. The proposed approach offers valuable insights useful for the advancement of morphing aircraft structures.

## 5. Conclusions

This study focuses on the design, optimization, and control of wing ribs with chordwise variable camber. The main conclusions drawn are as follows.

A complete design methodology for hybrid wing ribs has been introduced. This methodology initiates with the parametric representation of mean camber lines and the search for the optimal configuration. Subsequently, it employs the load path method for topology optimization, followed by an alternating shape/size optimization process based on subspace techniques. By integrating bionic designs, such as the ‘longbow’ beam and the ‘S’ structure, a structural solution that fulfills both strength and large deformation tolerance requirements is ultimately achieved.

The feasibility of pneumatic muscle actuators in such applications has been verified, and a high-performance controller matching them has been developed. Addressing the limitations of traditional PI sliding mode control in terms of parameter adaptability, the designed fuzzy adaptive PI sliding mode controller demonstrates excellent multi-condition control capabilities. Especially during large-angle deformations, it significantly reduces adjustment time and effectively suppresses overshoot.

The system performance was verified through experiments. The prototype of the flexible wing rib, made of 7075Al-T6 material, successfully achieved a target deflection of 15°, stress within the allowable range, and a deformation contour error of only 7.6%. Meanwhile, the control experiments confirmed that the system, under fuzzy adaptive control, exhibits comprehensive performance characterized by rapid response, small steady-state error, and low overshoot. Furthermore, the successful integration test of the three-rib wing box achieving a 30° downward deflection demonstrates the potential for the engineering applications of this research.

In summary, this study provides an effective technical path for achieving high-performance continuously variable camber airfoils, offering valuable insights for the future development of morphing aircraft.

## Figures and Tables

**Figure 1 biomimetics-11-00036-f001:**
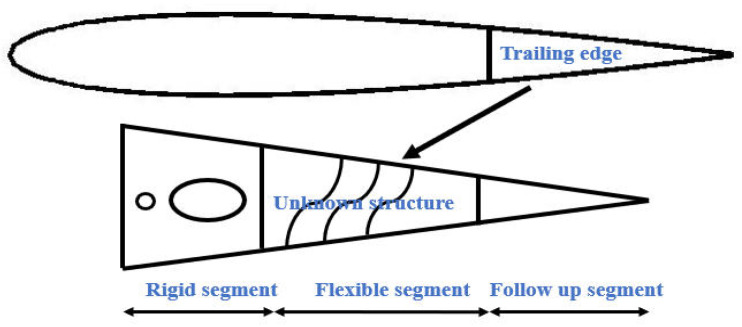
Rib trailing edge structure layout.

**Figure 2 biomimetics-11-00036-f002:**
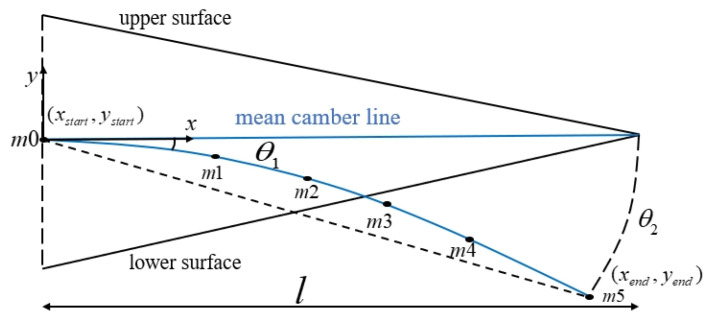
Mean camber line of airfoil trailing edge.

**Figure 3 biomimetics-11-00036-f003:**
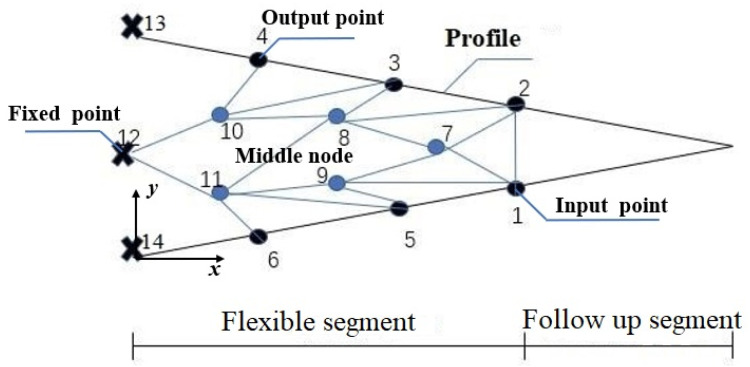
Topology optimization design region diagram.

**Figure 4 biomimetics-11-00036-f004:**
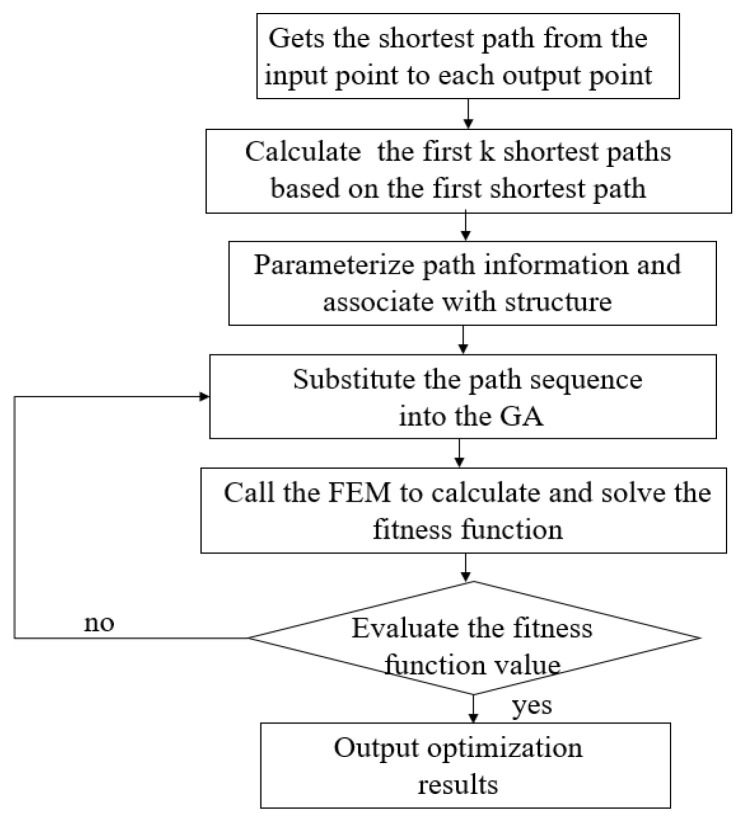
Topology optimization process based on load path method.

**Figure 5 biomimetics-11-00036-f005:**
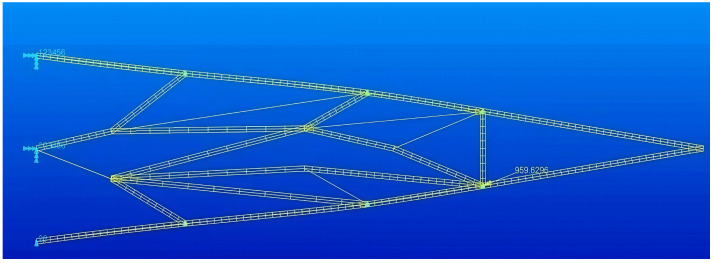
Topology optimization results of the flexible segment of the wing trailing edge.

**Figure 6 biomimetics-11-00036-f006:**
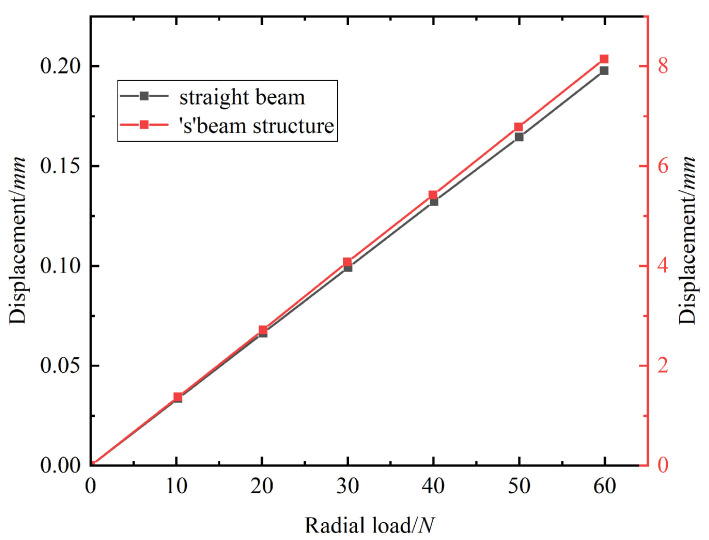
Load displacement curve of ‘S’ beam and straight beam.

**Figure 7 biomimetics-11-00036-f007:**
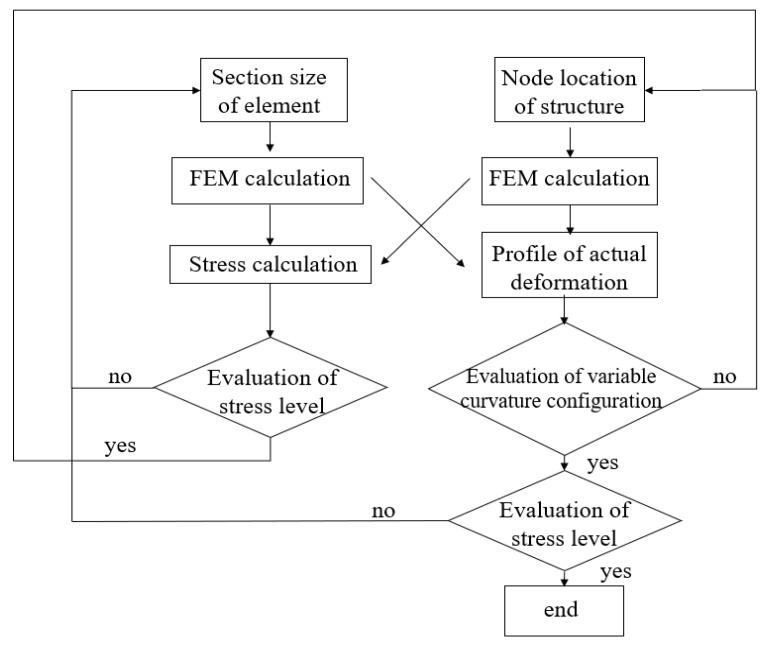
Subspace optimization iteration flowchart.

**Figure 8 biomimetics-11-00036-f008:**
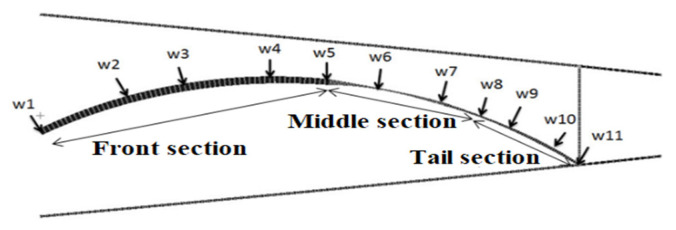
Dimension optimization of main through-beam node position.

**Figure 9 biomimetics-11-00036-f009:**
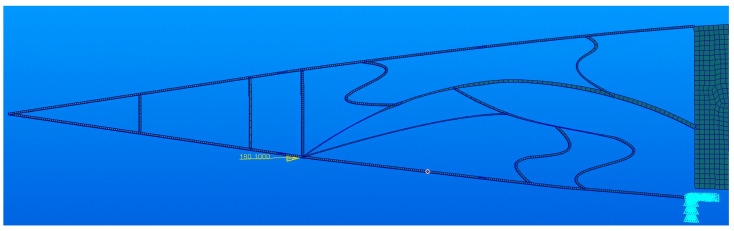
Finite element model of flexible wing rib structure.

**Figure 10 biomimetics-11-00036-f010:**
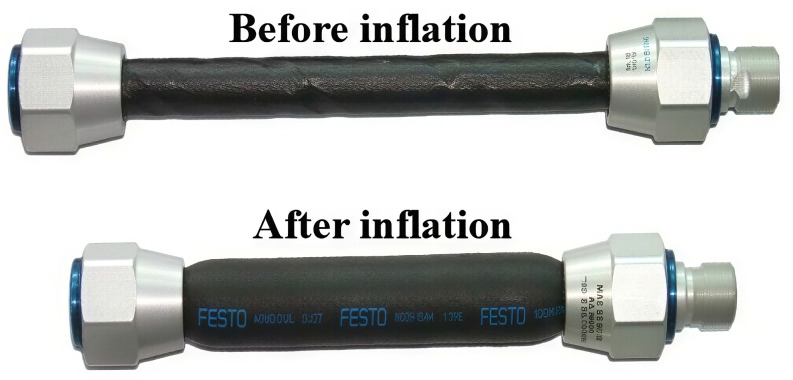
Schematic diagram of pneumatic muscle.

**Figure 11 biomimetics-11-00036-f011:**
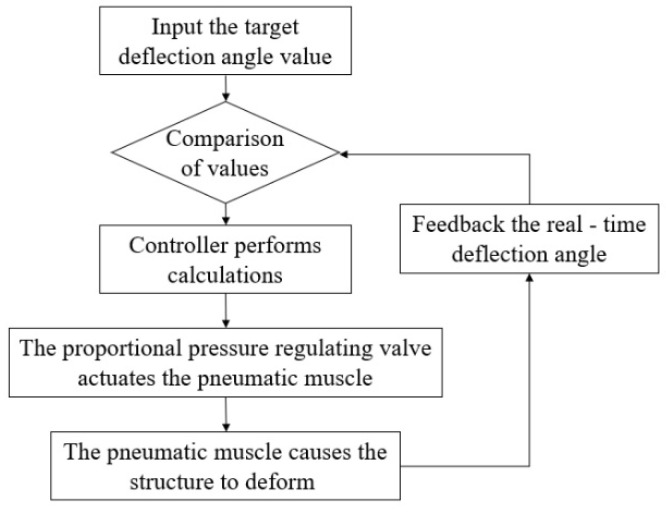
Flowchart of downward bias control in flexible segment.

**Figure 12 biomimetics-11-00036-f012:**
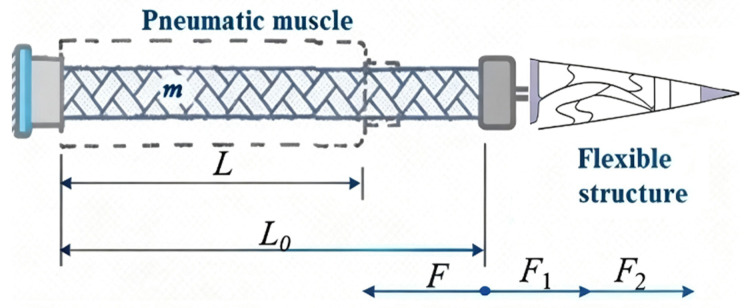
Pneumatic muscle force model diagram.

**Figure 13 biomimetics-11-00036-f013:**
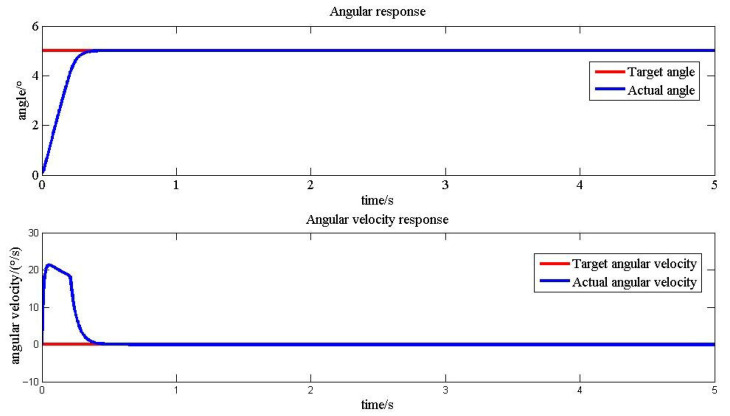
Simulation response curve corresponding to optimal control parameters.

**Figure 14 biomimetics-11-00036-f014:**
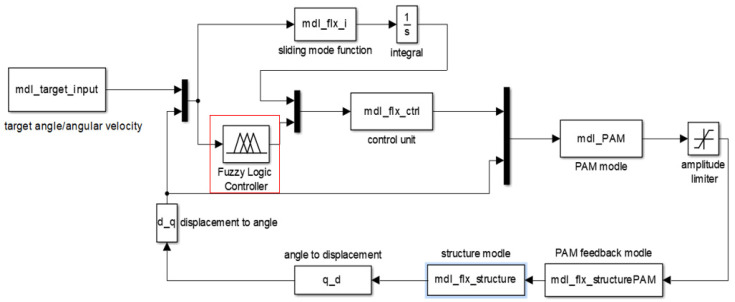
Drive control frame chart of control process simulation revised by fuzzy control.

**Figure 15 biomimetics-11-00036-f015:**
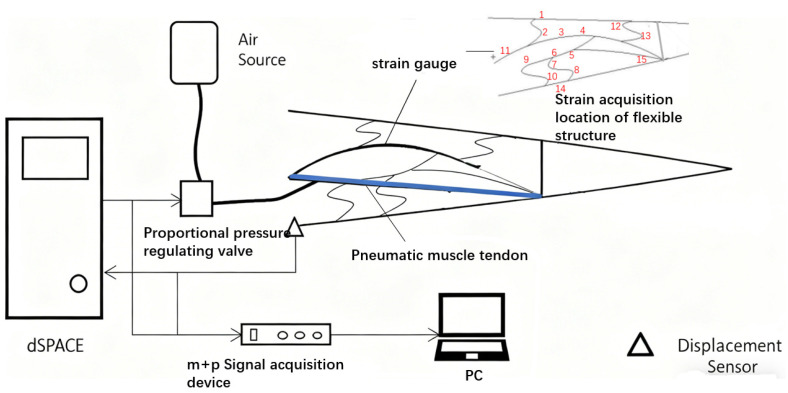
Schematic diagram of the experimental system.

**Figure 16 biomimetics-11-00036-f016:**
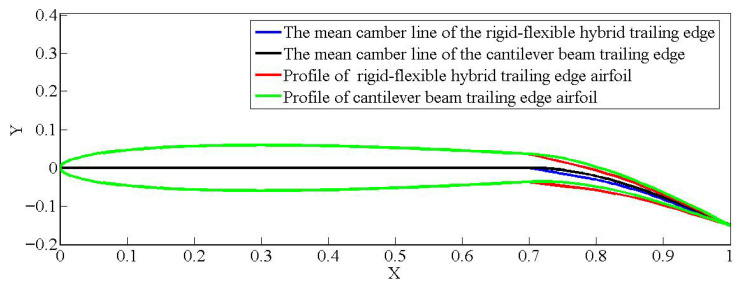
Optimal variable camber configuration of the flexible wing rib.

**Figure 17 biomimetics-11-00036-f017:**
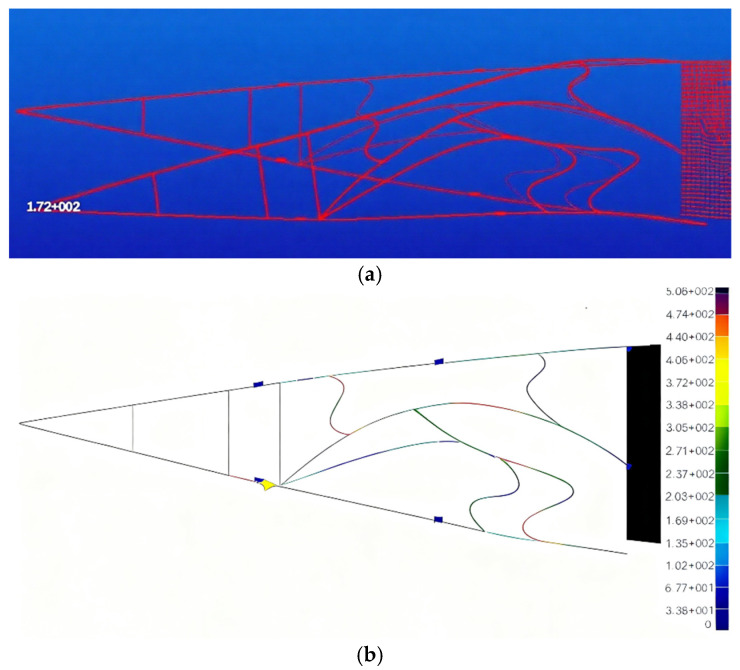
Variable bending configuration and stress distribution in the final generation: (**a**) variable curvature configuration of the final generation; (**b**) stress distribution of final generation.

**Figure 18 biomimetics-11-00036-f018:**
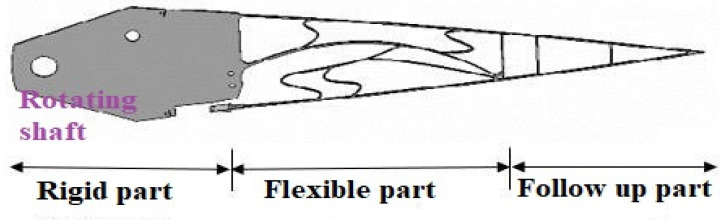
Optimal structure.

**Figure 19 biomimetics-11-00036-f019:**
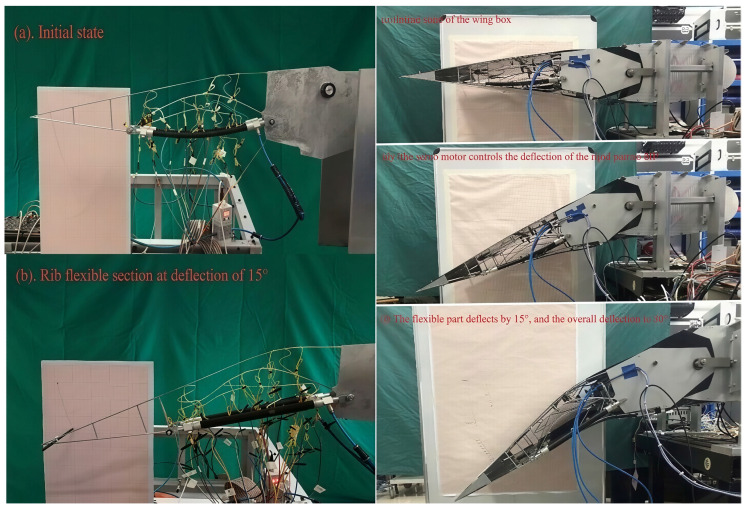
Photos of deformations in variable camber wing rib structure and wing box structure.

**Figure 20 biomimetics-11-00036-f020:**
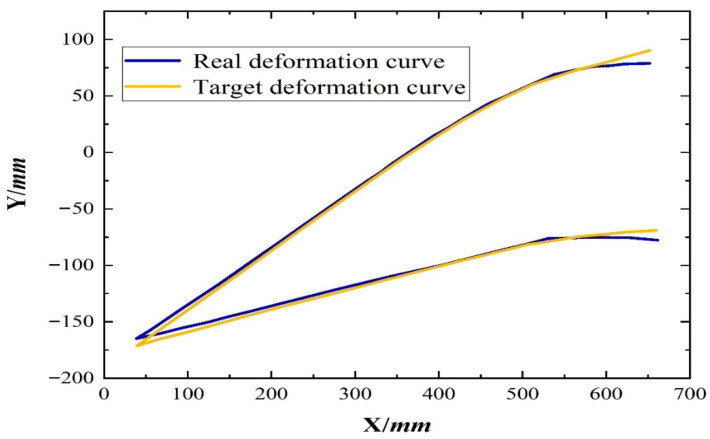
Comparison of optimal target deformation curve and real deformation curve.

**Figure 21 biomimetics-11-00036-f021:**
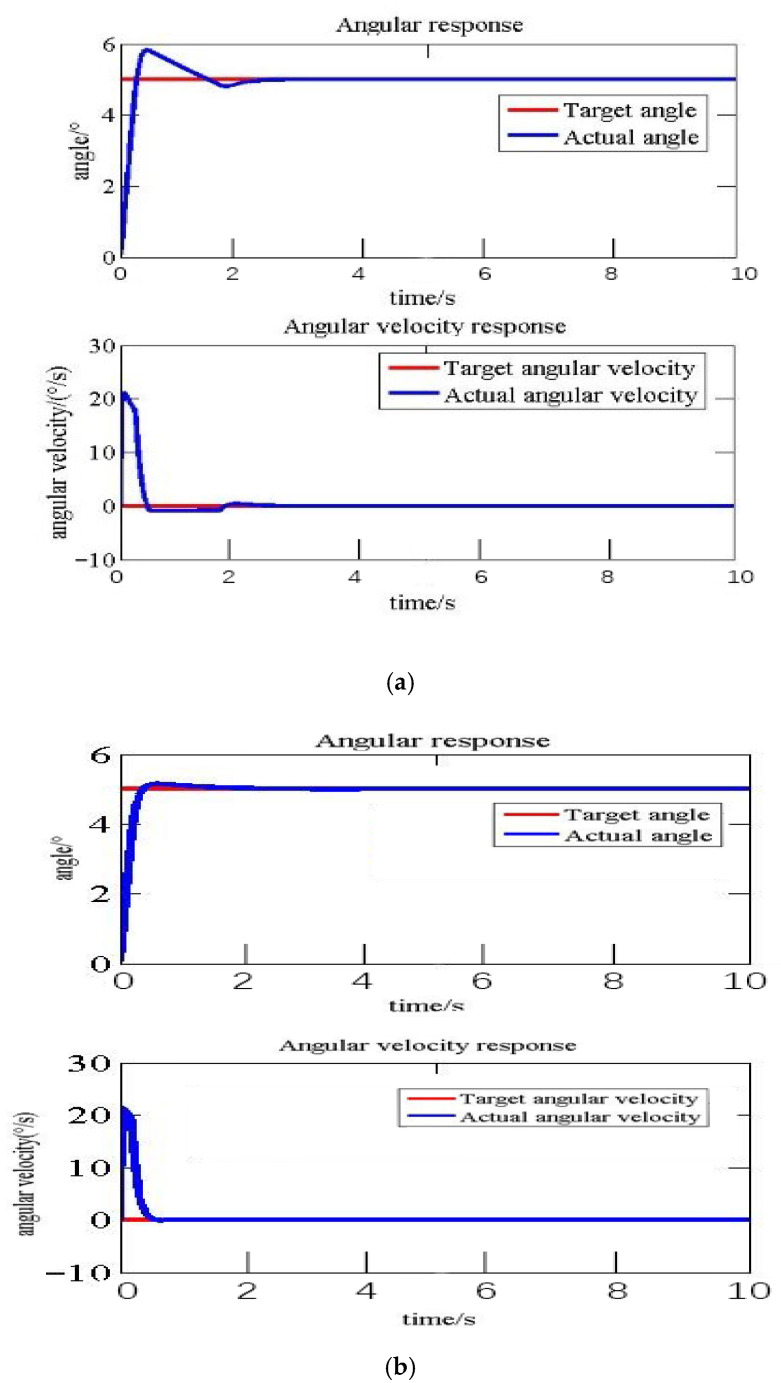
Comparison of fuzzy control response curves under non-optimal parameters: (**a**) $K_{p}=5,\;K_{i}=10,\;\Lambda =10$, without fuzzy control; (**b**) $K_{p}=5,\;K_{i}=10,\;\Lambda =10$, with fuzzy control.

**Figure 22 biomimetics-11-00036-f022:**
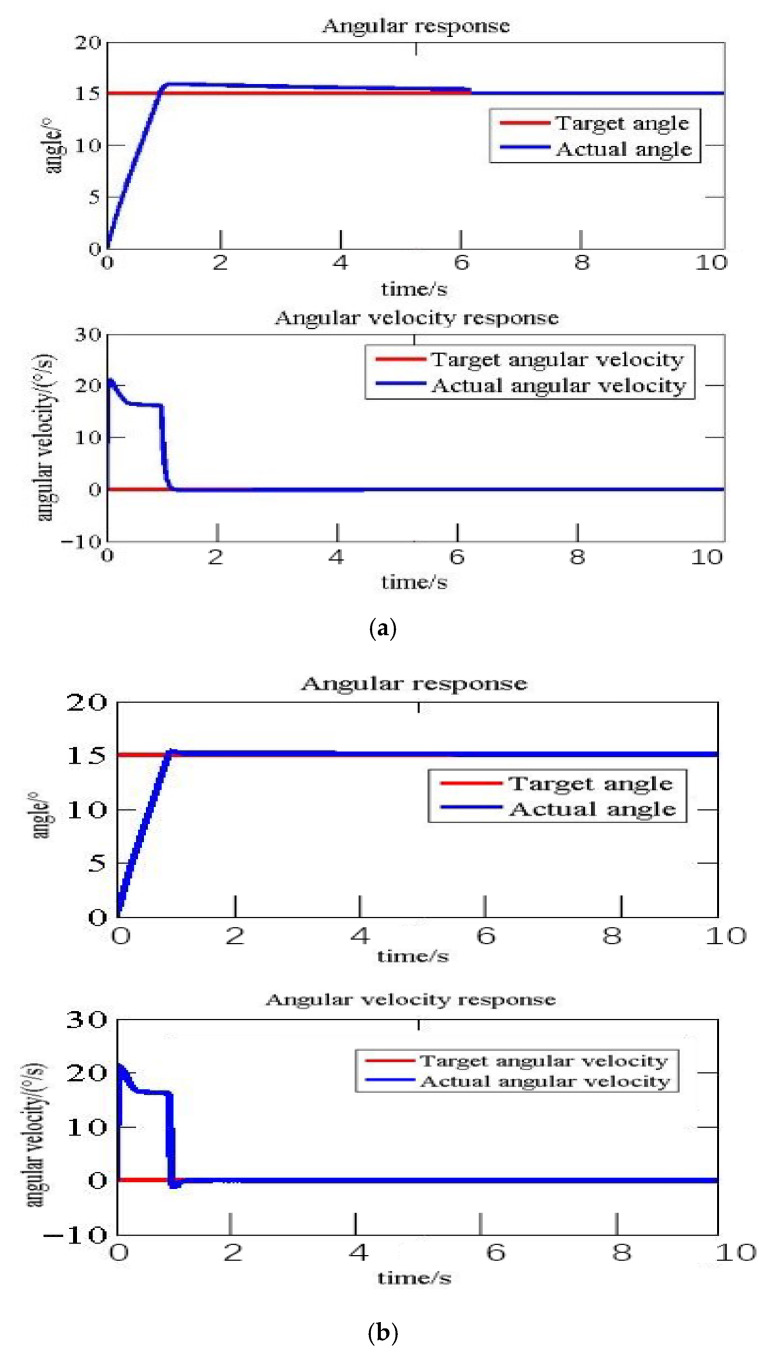
Comparison of fuzzy control response curve at maximum deflection angle: (**a**) $K_{p}=2,\;K_{i}=0.5,\;\Lambda =20$, without fuzzy control; (**b**) $K_{p}=2,\;K_{i}=0.5,\;\Lambda =20$, with fuzzy control.

**Figure 23 biomimetics-11-00036-f023:**
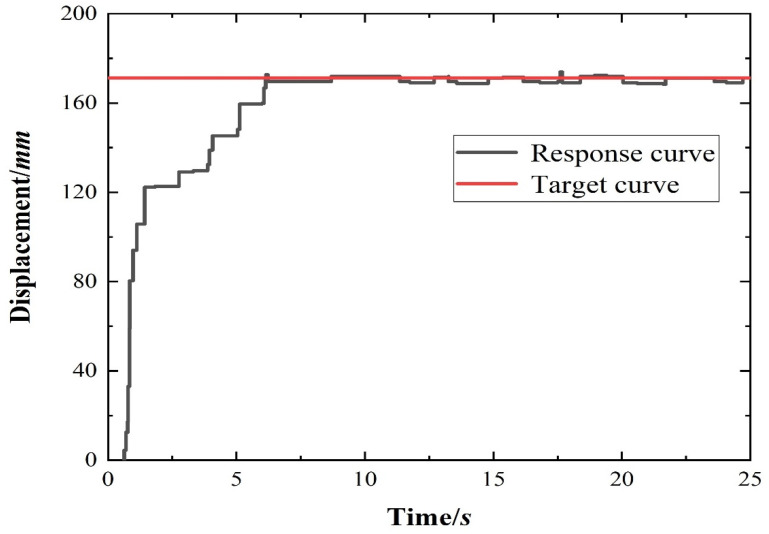
Fuzzy correction sliding mode control response curve for the flexible segment of the wing rib with variable camber.

**Table 1 biomimetics-11-00036-t001:** The airfoil state and environmental parameters.

AOA	Flow Velocity	Air Density	Viscosity Coefficient	Reynolds Number
8°	0.2 Ma	1.205 kg/m^3^	1.78938 × 10^−5^ Pa·s	4.58 × 10^6^

**Table 2 biomimetics-11-00036-t002:** The structure numbers in the first four paths from the input point to the output point.

Output Point/Path Sequence	First	Second	Third	Fourth
2	1,2	1,7,2	1,9,7,2	1,7,8,2
3	1,2,8,3	1,7,8,3	1,2,7,8,3	1,2,8,10,3
4	1,2,8,10,4	1,7,8,10,4	1,2,7,8,10,4	1,2,8,3,10,4
5	1,9,5	1,7,9,5	1,9,11,5	1,2,7,9,5
6	1,9,11,6	1,7,9,11,6	1,9,5,11,6	1,2,7,9,11,6

**Table 3 biomimetics-11-00036-t003:** Comparison of stability and deformation between the longbow beam and straight beam.

Structure Type	Section Size	Load (N)	Stability Factor	Endpoint Displacement (mm)
Longbow type	2.2 mm × 6 mm	(10,3)	9.75	4.17
Straight beam	2.2 mm × 6 mm	(10,3)	9.56	2.53

**Table 4 biomimetics-11-00036-t004:** Analysis Results of Grid Convergence.

Grid Size (mm)	Number of Elements	Maximum Stress (MPa)	Stress Change Rate (%)	Endpoint Displacement (mm)	Displacement Change Rate (%)
5	764	498.21		171.23	
3	1138	506.00	1.56	172.01	0.45
1	1542	508.52	0.49	172.46	0.26

**Table 5 biomimetics-11-00036-t005:** Structural parameters of the flexible wing rib.

Moment of Inertia	Weight	Centroid Position
0.012 kg × m^2^	0.096 kg	[0.293 0.004]

**Table 6 biomimetics-11-00036-t006:** Dynamic performance of the system under optimal control parameters.

*K_p_*	*K_i_*	Λ	Adjustment Time/s	Residual Error/%	Overshoot/%
2.0	0.5	20	0.34	0.03	0.00

**Table 7 biomimetics-11-00036-t007:** Fuzzy control rule table.

*ΔK_p_*, *ΔK_i_*		*e*				
		*NB*	*NS*	*ZO*	*PS*	*PB*
$\dot{e}$	*NB*	*PB NB*	*PB NB*	*PB NB*	*PS NS*	*ZO ZO*
	*NS*	*PB NB*	*PB NS*	*PS NS*	*PS ZO*	*NS PS*
	*ZO*	*PB NB*	*PS NS*	*ZO ZO*	*NS PS*	*NB PB*
	*PS*	*PS NB*	*ZO ZO*	*NS PS*	*NS PS*	*NB PB*
	*PB*	*ZO ZO*	*NB PS*	*NB PB*	*NB PB*	*NB PB*

**Table 8 biomimetics-11-00036-t008:** System’s dynamic performance under non-optimal control parameters.

Adjustment Mode	*K_p_*	*K_i_*	Λ	Adjustment Time (s)	Steady State Error (%)	Overshoot (%)
No fuzzy	5.0	10.0	10.0	3.11	0.00	16.41
Fuzzy	5.0	10.0	10.0	2.79	0.00	3.00

**Table 9 biomimetics-11-00036-t009:** Dynamic performance of the system during maximum angle deflection.

Adjustment Mode	*K_p_*	*K_i_*	Λ	Adjustment Time (s)	Steady State Error (%)	Overshoot (%)
No fuzzy	2.0	0.5	20	6.31	0.20	6.21
Fuzzy	2.0	0.5	20	1.23	0.19	2.33

**Table 10 biomimetics-11-00036-t010:** Control the relationship between pressure, root displacement, and deflection angle during the deformation process.

Driving Air Pressure	Displacement at the Root of the Wing (mm)	Deflection Angle (°)
0.00	0.00	0.00
0.81	9.08	3.01
1.56	26.15	6.08
2.35	43.23	9.06
3.14	56.14	12.03
3.91	69.06	15.01

## Data Availability

Data supporting the findings of this study are available from the corresponding author (B. Li, leebin@nwpu.edu.cn) upon reasonable request.

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
