# Peer review of "Research on Design and Control Method of Flexible Wing Ribs with Chordwise Variable Camber"

_biomimetics, 2026, doi:10.3390/biomimetics11010036_

Round 1

Reviewer 1 Report

Comments and Suggestions for Authors

  1. Table 3, please describe units in the parameters displayed.
  2. Figure6, the displacement curve for straight beam was flat. Please use secondary axis method to display the graph in better way.
  3. Please describe finite element model with mesh convergence and other boundary conditions sensitivity on results in details.
  4. Please also state the application of this type of wing morphing towards to aerospace products. How do this scale of prototype shown in experiments could be transformed to real application ?
  5. Please also give more details about the experiments and results data.

Author Response

Comment 1

Table 3, please describe units in the parameters displayed.

Response:

Thank you for this important observation. We have updated Table 3 to include clear units for all parameters. The revised table now explicitly states:

  • Load in Newtons (N)

  • Section size in millimeters (mm)

  • Endpoint displacement in millimeters (mm)

  • Stability factor as a dimensionless ratio

The updated table has been placed in the revised manuscript.

Comment 2

Figure6, the displacement curve for straight beam was flat. Please use secondary axis method to display the graph in better way.

Response:

We agree with the reviewer that the original presentation of Figure 6 did not effectively illustrate the differences between the two curves. As suggested, we have regenerated Figure 6 using a dual-Y-axis approach.

Comment 3

Please describe finite element model with mesh convergence and other boundary conditions sensitivity on results in details.

Response:

We thank the reviewer for emphasizing the importance of FEM validation. Within the article, we have provided additional details on finite element modeling, encompassing the material properties of the structure, the configuration of boundary conditions, and the validation of mesh convergence.

Mesh Convergence Analysis: A detailed table showing results from progressively refined mesh sizes, confirming convergence when the element count reached 1138 (changes in stress and displacement < 2%).

Comment 4

Please also state the application of this type of wing morphing towards to aerospace products. How do this scale of prototype shown in experiments could be transformed to real application ?

Response:

Thank you for suggesting this important perspective.The application of this type of variant aircraft is introduced in the introduction.

  • Commercial and transport aircraft: for improved lift-to-drag ratio and fuel efficiency.

  • High-maneuverability fighter aircraft: for enhanced agility, load alleviation, and stealth performance.

  • Next-generation morphing concepts: such as those under development in NASA's VCCTEF and Europe's Clean Sky programs.

Regarding the implementation approach, this paper presents the design and control methodologies for flexible wing rib structures. Presently, the sliding skin employed in the wing box test specimens remains unsealed at the chordwise endpoints. Consequently, a 3D-printed limiting structure is utilized for connection purposes. The research team is currently exploring the 'Ω' type periodic structure, with the objective of developing a flexible skin structure tailored for flexible wings. Concurrently, a load-bearing wing rib model has been fabricated using specialized steel. By simulating the aerodynamic loads equivalent to those experienced by a real medium-sized transport aircraft, static tests have been performed to confirm that the structure can sustain a favorable variable camber configuration under static loading conditions, with structural stress levels remaining within permissible limits. Looking ahead, the team plans to conduct wind tunnel tests and other relevant activities to enhance the technical maturity of this design.

Comment 5

Please also give more details about the experiments and results data.

Response:

Thank you for the suggestions provided by the reviewer.In the paper, we have elaborated on the equipment details of the experimental platform and the experimental procedure, encompassing the names and specifications of devices within the drive system, control system, and acquisition system. Furthermore, we have described the placement of the strain testing system. Regarding the experimental outcomes, we have incorporated data on the pneumatic muscle pressure during the control process, the displacement at the root of the wing rib's lower edge measured by the laser displacement sensor, and the real-time downward deflection angle.

We believe that the revised manuscript has been significantly improved thanks to the reviewers' constructive feedback. All changes have been highlighted in the revised document for ease of review. Should there be any further suggestions or required clarifications, we are pleased to incorporate them.

Once again, we express our sincere gratitude to the reviewers and the editorial team for your guidance and support.

Sincerely,
Xin Tao & Li Bin
Northwestern Polytechnical University

Reviewer 2 Report

Comments and Suggestions for Authors

The manuscript is an interesting contribution to the design and morphing of flexible wing ribs. Minor editorial corrections are needed. Otherwise, it can be published.

Author Response

Comment 

The manuscript is an interesting contribution to the design and morphing of flexible wing ribs. Minor editorial corrections are needed. Otherwise, it can be published.

Response:

We sincerely thank the reviewer for their positive and encouraging assessment of our work and for recommending our manuscript for publication. We are pleased to know that the reviewer finds our contribution interesting.

In light of the reviewer's recommendations, we have meticulously proofread the entire manuscript and implemented comprehensive editorial revisions. These revisions cover aspects such as grammar, spelling, formatting, reference citation style, as well as figures and tables.

We believe these editorial enhancements have further improved the readability and overall quality of the manuscript. All changes have been implemented in the revised version.

Once again, we express our gratitude to the reviewer for their valuable time and supportive comments.

Sincerely,
Xin Tao & Li Bin
Northwestern Polytechnical University

Reviewer 3 Report

Comments and Suggestions for Authors

Dear Authors,

The description of the method is clear, yet the tolerance of the measurement equipment should be noted.

It is good to see that both measurement and modelling were made.

The idea of figure 12 (and fig 19 and 20) raises few questions for me, first why was 5 second not more or less? If we assume every control mechanics that the first period is the relevant, then why haven’t you depicted the first second? Equations are not that complex in the method which would induce errors after the steady state is reached.

For me the rigid flexible word combination is contradicting. Maybe a semi-flexible or semi-rigid would be a better choice but it was just a suggestion.

Author Response

Comment 1

The description of the method is clear, yet the tolerance of the measurement equipment should be noted.

Response: 

We thank the reviewer for this valuable suggestion. We agree that specifying the measurement tolerances is crucial for assessing experimental accuracy. We have now included the accuracy specifications of key measuring equipment in the revised draft.

The measurement tolerance of the laser displacement sensor used for precise angle calculation is ±0.7% of the full scale. The strain coefficient tolerance of the strain gauge is ±1.0%. These tolerances were taken into account when analyzing the experimental results.

Comment 2

The idea of figure 12 (and fig 19 and 20) raises few questions for me, first why was 5 second not more or less? If we assume every control mechanics that the first period is the relevant, then why haven’t you depicted the first second? Equations are not that complex in the method which would induce errors after the steady state is reached.

Response:

We thank the reviewer for this valuable suggestion.The primary reason for selecting a 5-second time window in Figure 12 is that it clearly demonstrates that the system has reached a stable steady state, with no subsequent divergence or unexpected behavior, which is a crucial requirement for evaluating the performance of the control system. However, we fully agree with the reviewer's viewpoint that the control results in Figures 19 and 20 both reach steady state within 10 seconds. Therefore, we have adjusted the time range in Figures 19 and 20. This modification effectively highlights the transient phase, while still ultimately indicating that the system has reached steady state.

Comment 3

For me the rigid flexible word combination is contradicting. Maybe a semi-flexible or semi-rigid would be a better choice but it was just a suggestion.

Response:

We thank the reviewer for this thoughtful suggestion regarding terminology. We understand the point about the terms seeming contradictory.However, to avoid ambiguity and enhance the readability of the text, we have taken the following measures:

 We have retained the term "rigid-flexible coupled" only in the abstract and upon its first detailed definition in the introduction for consistency with established literature, but thereafter, we predominantly use simply "hybrid structure."

We believe that the revised manuscript has been significantly improved thanks to the reviewers' constructive feedback. All changes have been highlighted in the revised document for ease of review. Should there be any further suggestions or required clarifications, we are pleased to incorporate them.

Once again, we express our sincere gratitude to the reviewers and the editorial team for your guidance and support.

Sincerely,
Xin Tao & Li Bin
Northwestern Polytechnical University

Reviewer 4 Report

Comments and Suggestions for Authors

The topic is well introduced and explained, all steps (optimal configuration - topology optimization - subspaces-based optimization) are thoroughly reasoned and described. The results are convincing, the precision and overshoot of the fuzzy-modified PI-SMC are very good. There is also a good agreement between the simulations and experiment.

Overall, this paper is indeed a viable framework for the topic of chordwise morphing wings and can serve for future research considering aeroelasticity and and other experiments. The quality is very high, I only noticed a few small mistakes in the text:

Line 203: The sentence "date the coordinates in the optimization variables using genetic algorithms." seems to be incomplete.

Line 485: There should rather be "A test system with pneumatic muscles … was built" instead of "Built a test system ..."

Line 501: The text says "the rigid-flex hybrid trailing edge demonstrates … smoother profile compared to the cantilever beam trailing edge.", yet visually, the Figure 14 seems to suggest otherwise (or maybe it is just unclear what is meant by "smoother").

Author Response

We are very grateful to the reviewer for their thorough review and highly positive assessment of our work.We are also thankful for the reviewer's keen eye in identifying the minor textual errors, which we have corrected as detailed below.

Comment 1

Line 203: The sentence "date the coordinates in the optimization variables using genetic algorithms." seems to be incomplete.

Response:

We thank the reviewer for pointing out this error. The word "date" was a typographical error. The intended verb was "update". The sentence has been corrected in the revised manuscript to read:

"Update the coordinates in the optimization variables using genetic algorithms."

Comment 2

Line 485: There should rather be "A test system with pneumatic muscles … was built" instead of "Built a test system ..."

Response:

We agree with the reviewer. The original sentence was grammatically incorrect as it was a fragment. We have rewritten the sentence for better clarity and grammatical accuracy. The revised text now reads:

"A test system consisting of drive, control, and measurement modules was constructed; a 15° downward bending test was conducted, and the adjustment time, steady-state error, and overshoot were recorded."

Comment 3

Line 501: The text says "the rigid-flex hybrid trailing edge demonstrates … smoother profile compared to the cantilever beam trailing edge.", yet visually, the Figure 14 seems to suggest otherwise (or maybe it is just unclear what is meant by "smoother").

Response:

We extend our heartfelt thanks to the reviewer for their keen observation and for pointing out the errors. The phrase "more smooth" was indeed used imprecisely; the author's intent was to assess whether the curvature distribution along the mean arc was more uniform, thereby potentially enhancing aerodynamic performance. Cantilever beam deformation typically leads to a concentration of curvature near the root section.

To eliminate any potential ambiguity, we have refined the text in the manuscript to convey this point with greater precision: "As illustrated in the figure, when subjected to an identical equivalent downward bending angle, the rigid-flex hybrid trailing edge demonstrates a more pronounced downward deflection and a more seamless curvature distribution compared to the trailing edge of the cantilever beam."

We believe that the revised manuscript has been significantly improved thanks to the reviewers' constructive feedback. All changes have been highlighted in the revised document for ease of review. Should there be any further suggestions or required clarifications, we are pleased to incorporate them.

Once again, we express our sincere gratitude to the reviewers and the editorial team for your guidance and support.

Sincerely,
Xin Tao & Li Bin
Northwestern Polytechnical University